# Optical inter-site spin transfer probed by energy and spin-resolved transient absorption spectroscopy

Felix Willems [1], Clemens von Korff Schmising [1✉], Christian Strüber [1], Daniel Schick [1], Dieter W. Engel [1], J.K. Dewhurst[2], Peter Elliott[1], Sangeeta Sharma[1] & Stefan Eisebitt[1,3]

Optically driven spin transport is the fastest and most efficient process to manipulate macroscopic magnetization as it does not rely on secondary mechanisms to dissipate angular momentum. In the present work, we show that such an optical inter-site spin transfer (OISTR) from Pt to Co emerges as a dominant mechanism governing the ultrafast magnetization dynamics of a CoPt alloy. To demonstrate this, we perform a joint theoretical and experimental investigation to determine the transient changes of the helicity dependent absorption in the extreme ultraviolet spectral range. We show that the helicity dependent absorption is directly related to changes of the transient spin-split density of states, allowing us to link the origin of OISTR to the available minority states above the Fermi level. This makes OISTR a general phenomenon in optical manipulation of multi-component magnetic systems.

[1] Max Born Institute for Nonlinear Optics and Short Pulse Spectroscopy, Max-Born-Strasse 2A, 12489 Berlin, Germany. [2] Max-Planck-Institute for Microstructure Physics, Weinberg 2, 06120 Halle (Saale), Germany. [3] Institut für Optik und Atomare Physik, Technische Universität Berlin, 10623 Berlin, Germany. ✉email: korff@mbi-berlin.de

Ultrafast magnetism belongs to one of the most active fields in solid-state physics with frequent discoveries of new fundamental microscopic phenomena. Prominent examples include all-optical magnetization switching mediated by a transient ferromagnetic state[1,2], control of antiferromagnetic order[3–7] and transfer of angular momentum via terahertz spin currents[8–13]. Very recently it was shown theoretically, that in multi-component magnetic systems laser-driven optical transitions can induce a spin-selective charge flow between sub-lattices causing significant magnetization changes, including switching from antiferromagnetic to ferromagnetic order[14–18]. Indeed, a first experiment employing attosecond spectroscopy in the extreme ultraviolet spectral range (XUV) was able to demonstrate how a light-field driven coherent charge relocation leads to an ultrafast loss of magnetization[19]. Since such optically induced spin transfer is most pronounced during the pulse, it represents the fastest mechanism to control magnetization with light, promising applications for future data storage and spintronic devices.

Nonetheless, the experimental detection of the presence of OISTR in a given material remains challenging, as it requires access to the element-specific transient density of states (DOS) around the Fermi energy, $E_F$. While time-resolved photoelectron spectroscopy can access the valence band structure of magnetic solids in great detail[20–23], it is limited to surfaces and does not provide element specificity. To gain access to ultrafast changes of element-specific magnetization, an increasing number of experiments have made use of the magnetic contrast at $3p$ resonances in the XUV spectral range. It was demonstrated that this technique gives access to the microscopic interplay of different elements in multi-component magnetic systems in a single measurement[24–26] and allows distinguishing laser-driven local and non-local magnetization dynamics[12,27] as well as exchange[28] and collective spin excitations[29,30]. In particular, magnetic circular dichroism (MCD) in transmission geometry[19,25] can be explained by a simple two-step model[31]: absorption of circularly polarized photons in resonance with a core to valence state transition leads to spin-polarized photoelectrons via spin-orbit (SO) interaction. The different number of available states in the majority and minority channels in the exchange-split valence band then determines the

energy and helicity dependent absorption strength in the magnetic material. The typical observable in such experiments is the magnetic asymmetry, defined as the normalized difference of the transmitted intensities for left and right circularly polarized photons, $A = (I_{\sigma+} - I_{\sigma-})/(I_{\sigma+} + I_{\sigma-})$. Hence, time-resolved XUV MCD does not give direct information about the energy-dependent electronic state distribution, which is required for monitoring OISTR as a function of time.

In this letter, we demonstrate that the analysis of the helicity and energy-dependent absorption, $\mu_{\sigma\pm} = 1 - I_{\sigma\pm}$, as a function of delay between pump and probe pulses reveals detailed information on the transient spin-split DOS, which we can directly compare to theoretical simulations. We employ helicity dependent absorption spectroscopy around the $M_{2,3}$ resonance of Co and compare the response of a pure Co film and a CoPt alloy to demonstrate that optical inter-site spin transfer (OISTR) between Pt and Co atoms in the two-component system CoPt dominates the early time dynamics leading to a more efficient and faster demagnetization. Furthermore, we highlight the importance of the dynamics of minority spins in physics of ultrafast demagnetization, a contribution usually ignored in most modern-day descriptions of femtomagnetism[15,16,32].

## Results

**Experimental details.** In the time, energy and spin-resolved XUV measurements, we recorded the transmitted spectral intensity, $I_{\sigma\pm}$, of circularly polarized higher harmonic radiation through the magnetic sample as a function of the time delay, $\Delta t$, between the pump (center wavelength 800 nm, pulse duration 60 fs) and the higher harmonic probe pulses (cf. Fig. 1a). The higher harmonic radiation is generated by focusing intense laser pulses with an energy of 2.5 mJ, a pulse duration of 30 fs and a photon energy centered at $E_{Ph} = 1.55$ eV into a neon-filled gas cell leading to a spectrum of discrete harmonic emission peaks separated by $2E_{Ph}$ (cf. Fig. 1c). The width of one harmonic peak is approximately 200 meV. A 4-mirror phase shifter[25,33,34] is used to control the state of polarization. We switch the magnetization of the sample by an external magnetic field, equivalent to changing the helicity

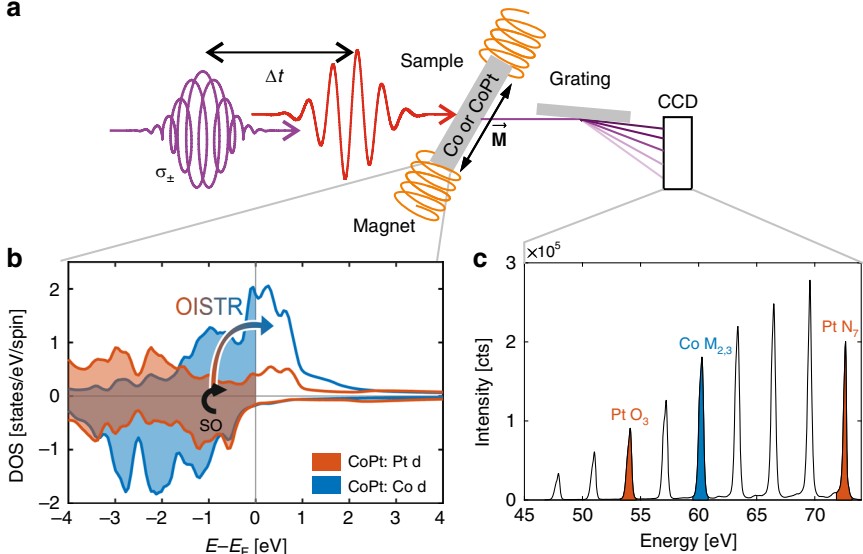

**Fig. 1 Experimental set-up and microscopic mechanism of OISTR in CoPt. a** Experimental geometry showing the optical excitation pulse as well as the time-delayed, circularly polarized XUV pulse (σ±). The XUV pulse is transmitted through the sample, energetically dispersed by a grating and then detected by a CCD camera. **b** Spin-dependent $d$-DOS of Co and Pt of the $Co_{50}Pt_{50}$ alloy. The high spin-orbit coupling on Pt $5d$ leads to an enhanced spin-flip rate localized on the Pt atom (black arrow). The colored arrow denotes the OISTR process; minority carriers are optically excited from Pt $5d$ to Co $3d$ states. **c** The measured high harmonic spectrum with indicated resonances for Pt $O_3$ and $N_7$ and Co $M_{2,3}$ transitions.

of the circularly polarized photons, σ±[35]. The full XUV spectrum transmitted through the sample is energetically dispersed and measured with an XUV-sensitive charge-coupled device, CCD. We use a second spectrometer before the sample to measure the incoming higher harmonic spectrum and to normalize the intrinsic fluctuations in the higher harmonic intensities, which results in an increased signal to noise ratio. Both spectrometers are set to resolve the spectral range from 47 to 73 eV and we record nine harmonic emission peaks simultaneously. The joint exposure time is set between 100 ms and 500 ms. We repeat a delay scan measurement up to 100 times, resulting in a total measuring time of approximately 4 h for each sample. The resulting root mean square of the measured signals remains well below $5 \times 10^{-4}$ allowing us resolving details of the ultrafast changes of small magnetic asymmetries below $1 \times 10^{-2}$.

The samples, elemental Co and a $Co_{0.5}Pt_{0.5}$ alloy, were grown by DC magnetron sputtering with a thickness of 15 nm on freestanding 20 nm thick $Si_3N_4$ and 200 nm thick Al membranes, respectively. In both cases, electronic conduction between the magnetic layer and the substrate can be excluded, as the surface of the Al-membrane is oxidized before deposition. As we investigate the samples in transmission geometry, contributions to a local demagnetization by superdiffusive spin transport do not play any role. The samples are magnetized in-plane and consequently mounted under a grazing angle of 45 degrees for a finite projection of the **k**-vector of the circularly polarized XUV pulses onto the magnetization of the sample. Both samples were excited with the same incident fluence of 12 mJ/cm² in a collinear pump-probe geometry.

**Theoretical details**. The simulations are performed by employing a two-step process: first, density functional theory (DFT) is used to obtain the electronic ground-state of the material. Second, we study the dynamics of the charge and the magnetization density under the influence of an external laser pulse by using the time-dependent extension of density functional theory (TD-DFT). The whole procedure is fully ab initio and at no point requires any input from experiments, except for the pump-pulse parameters (fluence, pulse duration, and frequency). The simulations are performed using a fully non-collinear version of TD-DFT as implemented within the Elk code[46], where the dynamics of the laser-excited charges and spins are treated beyond the linear-response regime and by including relativistic effects (see Methods for further details).

Thus, the following mechanisms are included in the theoretical simulations: (i) spin and charge currents due to laser excitation; this term is responsible for the OISTR mechanism, (ii) SO induced spin-flips, (iii) effect of these spin-flips on the spin-currents[16], (iv) electron-electron scattering effects via the exchange-correlation functional, and (v) coupling of the electrons to nuclear degrees of freedom; the change in charge density as a function of time (due to laser excitation) leads to large forces on the nuclei which are coupled to the spins via Coulomb and SO terms. Second-order effects are missing in our simulations—these forces displace nuclei (generate phonons) and this, in turn, causes charge and spin density to change. However, we note that such second-order effects start to cause significant changes to magnetization only after ~100 fs, and hence, we have focused on the early time physics in the present work.

**Ultrafast demagnetization**. In Fig. 2a, we show the temporal evolution of the relative magnetic asymmetry, A(t), for selected photon energies. For the Co film, we show the response at the Co $M_{2,3}$ resonances at 60.3 eV, and for the CoPt alloy, the responses at the Co $M_{2,3}$ (60.3 eV) as well as for the Pt $O_3$ (54.1 eV) and Pt

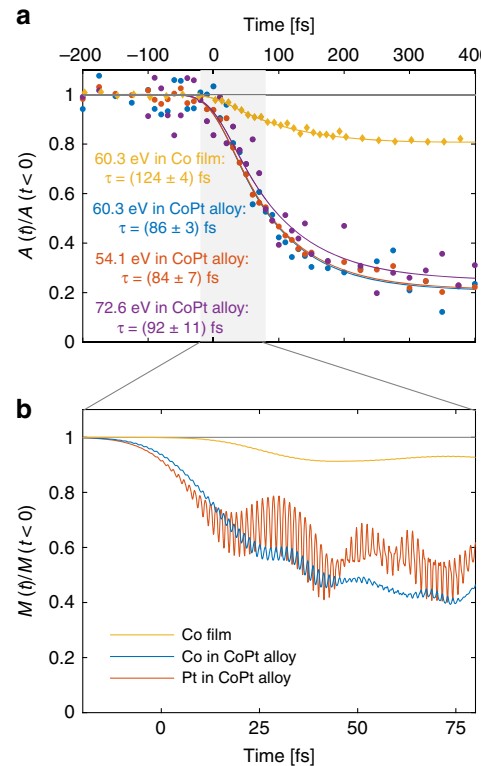

**Fig. 2 Ultrafast demagnetization after optical excitation. a** Measured relative magnetic asymmetry for a Co film and a $Co_{0.5}Pt_{0.5}$ alloy as a function of pump-probe delay. The demagnetization of the CoPt alloy exhibits a larger amplitude and faster rate compared to the pure Co film. In CoPt, the responses at 60.3, 54.1, and 72.6 eV corresponding to the Co $M_{2,3}$ edge and Pt $O_3$ and $N_7$ edges are all identical. **b** Calculated change of the magnetization for a pure Co film as well as for the elements Co and Pt in a $Co_{0.5}Pt_{0.5}$ alloy. Note that the calculations only show the very early dynamics up to 80 fs (shaded area in (**a**)).

$N_7$ (72.6 eV) resonances are shown. After ~400 fs the magnetic asymmetry of the Co film at the Co $M_{2,3}$ edge is reduced by 20%. The maximal reduction of the asymmetry for the CoPt alloy is significantly larger and amounts to 80% for all probe photon energies. The solid lines are non-linear least square fits (see the "Methods" section and Supplementary Information for details). Within the experimental uncertainty, the element-specific demagnetization times in the CoPt alloy are comparable for both photon energies, in particular, we find a value of $\tau = (86 \pm 3)$ fs and $\tau = (84 \pm 7)$ fs at 60.3 eV and 54.1 eV, respectively. Note that because of a non-negligible magnetic asymmetry of Co for photon energies well below its resonance[36] and the broad spectral signature of the Pt O edges, we can only ascribe a predominant sensitivity for Co at 60.3 eV and a predominant sensitivity for Pt at 54.1 eV. However, data at the Pt $N_7$ edge also exhibits—within the experimental uncertainty—an identical time constant of $\tau = (92 \pm 11)$ fs as well as an equal demagnetization amplitude of 80%. A comparable demagnetization time and amplitude for Co and Pt in the CoPt alloy is also predicted by theoretical calculations as shown in Fig. 2b (a detailed discussion on the origin of the observed oscillation is given in the Methods section). However, comparison of the demagnetization amplitude and time constant at the Co $M_{2,3}$ edges between the pure Co film and the CoPt alloy yields significant differences, both in experiment and theory. We measure $(124 \pm 4)$ fs in the Co film and $(86 \pm 3)$ fs in the CoPt alloy. Shorter demagnetization times in the presence of heavy elements like Pt or Pd have been observed previously[37] and

have been attributed to an increased spin-flip probability during electron scattering events mediated by larger SO interaction[38].

What is novel in our work is that we can directly follow the element-specific magnetization dynamics of the two constituents of the CoPt alloy. The experimental observation of identical demagnetization rates for Co and Pt is unexpected, given the difference in their respective SO coupling strengths; the SO coupling strength is approximately 10 times larger in Pt than Co[39]. This can be qualitatively understood by recognizing that the SO coupling is proportional to the gradient of the local-potential (see Eq. 1 in "Methods" section). This implies that, first, it is only large in the region around each atom[15] and, second, it should scale with the nuclear charge $Z$ (it can be shown that for outer $d$-electrons it scales according to $Z^2$[39]). In other words, a higher SO coupling should entail a larger local demagnetization rate for heavy elements as compared to transition metals in multi-component magnets.

This is in contrast to our experimental observations and theoretical calculations. In the following, we will show that this unexpected behavior is due to the mechanism of OISTR, which plays an additional important role and leads to an equal demagnetization rate in both species. To corroborate this, we first look in detail at the measured and calculated energy and helicity dependent absorption.

**Helicity dependent transient absorption**. In Fig. 3a, we plot the experimental data for the normalized helicity dependent absorption, $\mu_{\sigma\pm}$, as a function of pump-probe delay for pure Co and the CoPt alloy at the Co $M_{2,3}$ resonances at 60.3 eV. We observe that $\mu_{\sigma+}$ increases for both samples on a similar timescale after laser excitation. We attribute this to an increase of available states driven by laser excitation, predominantly in the majority channel. The dynamics of $\mu_{\sigma-}$, however, show a strikingly different behavior: while for the CoPt sample, it decreases without delay, in the pure Co film the absorption remains constant for approximately 60 fs before it starts to drop as well. Analogously, we understand a decrease of $\mu_{\sigma-}$ by filling of previously available states, now mainly in the minority channel. Hence, the direct comparison of the transient absorption data between Co and CoPt data gives a strong indication that an inter-site redistribution of minority spin electrons plays an essential role in the enhanced demagnetization in CoPt alloy at early times.

In order to understand the difference between Co and CoPt, we first theoretically calculate the same observable as in the experiment, namely the transient helicity dependent absorption. This is done by performing linear-response calculations using the laser-excited state of the system at various times following the procedure detailed in Dewhurst et al.[40]. Figure 3b displays the theoretical helicity dependent absorption at 60.3 eV for Co and CoPt. For both samples we observe an increase of $\mu_{\sigma+}$, more pronounced for Co than for CoPt in very good agreement with the experimental data. The transient absorption, $\mu_{\sigma-}$, for Co stays approximately constant before it starts to drop after about 60 fs. The calculation also reproduces the very pronounced decrease of $\mu_{\sigma-}$ for CoPt. Note that the calculations only give a reliable description of the underlying microscopic physics for early times after excitation (marked as a gray shaded area in Fig. 3a), as the theory does not contain the second-order coupling to phonons and radiative effects (as discussed previously) which start to dominate after approximately 100 fs.

**Transient density of states**. To gain an in-depth understanding of the distinct dynamics of the majority and minority spin as implied by the helicity dependent absorption, we calculate the transient DOS using TD-DFT. The results are displayed in Fig. 3c

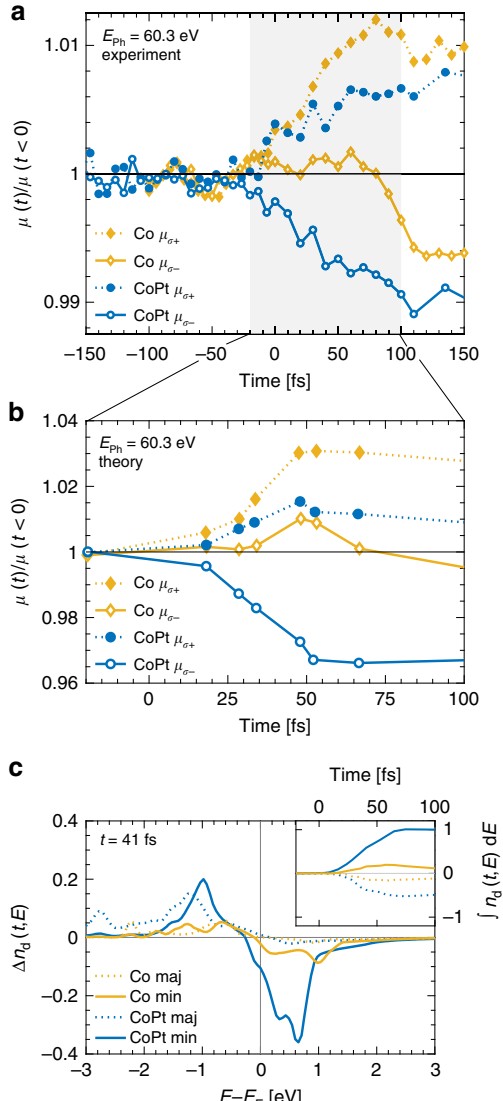

**Fig. 3 Helicity-dependent transient absorption and corresponding evolution of the density of states. a** Experimentally measured and **b** theoretically calculated normalized helicity dependent absorption, $\mu_{\sigma\pm}$, for the Co and CoPt sample at the Co $M_{2,3}$ resonance at 60.3 eV as a function of pump-probe delay. We interpret the qualitative difference of the response of $\mu_{\sigma-}$ between the single and two-component samples as an efficient filling of available minority states above the Fermi level, $E_F$, in CoPt. Note the excellent agreement between experiment and theory. **c** The calculated difference of occupied minority and majority $d$-states, $\Delta n(t = 41$ fs, $E)$ after optical excitation for Co and CoPt. Laser excitation leads to an increase of available states, $\Delta n(t, E)$, below, and a decrease above $E_F$, the latter being strongly enhanced for CoPt. The inset compares the energy-integrated occupied Co $d$-states, $n_d$, for the Co and CoPt samples. It clearly shows that in CoPt, the increase of minority electrons exceeds the loss of majority electrons. This is explained by OISTR transferring minority electrons from Pt to Co.

where we plot the difference of occupied $d$-states between the ground-state and the excited system, $\Delta n_d(t, E) = n_d(t < 0, E) - n_d(t, E)$, as a function of energy for a selected time ($t = 41$ fs) after optical excitation. The laser excitation results in an increase of available states below $E_F$, i.e., $\Delta n_d(t > 0, E < E_F) > 0$ and filling of previously empty states above $E_F$, i.e., $\Delta n_d(t > 0, E > E_F) < 0$. Closer inspection of majority spin electrons for Co as well as for CoPt reveals that the gain of available states below $E_F$ does not

correspond to the loss of available states above $E_F$, i.e., $\Delta n_d(t = 41$ fs, $E < E_F) > \Delta n_d(t = 41$ fs, $E > E_F)$. This is equivalent to saying that there is a net loss of majority spin electrons, leading to an overall increase of $\mu_{\sigma+}$. Since the total charge is conserved, spin-flip processes from majority to minority states are a rational explanation for this observation[15,38,41].

However, surprisingly, we observe that the evolution of the minority states above $E_F$ in Co and CoPt shows a qualitative difference despite identical laser excitation. In the Co film, the changes of available minority electrons below and above the Fermi energy are approximately equal resulting in an unchanged absorption $\mu_{\sigma-}$. Only after about 60 fs with progressing electron redistribution, spin-flip processes, i.e., an increase of minority electrons, are directly evidenced by a decrease of $\mu_{\sigma-}$. Very differently, in CoPt the minority states above $E_F$ are filled much more efficiently immediately with the laser excitation leading to a rapid decrease of available states and a corresponding reduction of $\mu_{\sigma-}$.

One can appreciate the underlying microscopic origin by looking at the inset of Fig. 3c, where we plot the number of occupied spin-split $d$-states, $n_d$, integrated in the energy range $-13$ eV to $+13$ eV. Minority electrons of Co in CoPt (solid blue line) show a large increase, significantly exceeding the loss of majority electrons for Co in CoPt (dotted blue line). In contrast, in the Co film, the increase in minority electrons is equal to the loss of majority electrons (dotted and solid yellow line respectively). This demonstrates that in CoPt the demagnetization is not only driven by an increased local spin-flip rate, which would entail an equal change in the majority and minority spin electrons—i.e., a loss in majority is equal to the gain in minority for Co in CoPt. Instead, the integration over all energies shows that there is an additional source of minority electrons flowing to Co $3d$ states in CoPt. Since the total charge is conserved and the contribution of $s$ and $p$ states to the dynamics was found to be insignificant in our theoretical simulations, the net gain of minority electrons localized at Co must come from the Pt $5d$ states. This is corroborated by a detailed analysis of the Pt $5d$ charge dynamics shown in the Supplementary Fig. 2.

Hence, the analysis of the transient DOS gives strong evidence of an optically driven spin transfer (OISTR) from Pt $5d$ states to Co $3d$ states, responsible for very efficient demagnetization during the presence of the optical laser excitation pulse. Finally, the theoretical observation that the source of minority electrons are Pt $5d$ states is experimentally reaffirmed by helicity dependent absorption measurements at the Pt $N_7$ edge at 72.6 eV shown in Fig. 4. Again, we ascribe relative changes in the absorption of $\sigma_-$ and $\sigma_+$ light to changes predominantly in the minority and majority state occupations, respectively. Accordingly, we interpret the strong increase of $\mu_{\sigma-}$ as a loss of minority electrons at the Pt site. This further supports our hypothesis that the source of additional minority electrons in Co in the CoPt alloys is indeed Pt.

We summarize the two essential processes that lead to enhanced demagnetization in CoPt in Fig. 1b: a large SO coupling on the Pt site causes an increased spin-flip rate (black arrow) while laser excitation leads to an efficient transfer of minority electrons from Pt to Co via OISTR (colored arrow).

Our experimental data, together with the ab initio calculations, show that energy and helicity dependent transient absorption (HDXAS) gives access to spin-resolved transient changes of the DOS around the Fermi level of a specific element. This observable gives clear evidence that in a CoPt alloy two processes lead to an increased demagnetization amplitude and rate: enhanced spin-flips mediated by SO coupling and optically driven spin transport from Pt $5d$ minority electrons to the Co $3d$ states. The driver for the flow of minority carriers is found in the large phase space of

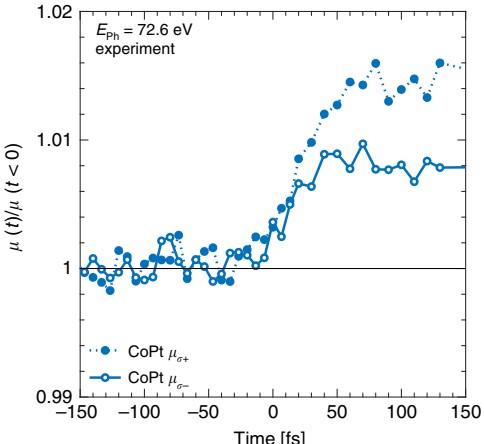

**Fig. 4 Helicity dependent transient absorption at the Pt $N_7$ edge.** Measured helicity dependent absorption, $\mu_{\sigma\pm}$, at the Pt $N_7$ edge at 72.6 eV. Relative changes in the absorption of light with $\sigma_-$ and $\sigma_+$ helicity is predominantly caused by available states in the minority and majority states, respectively. The strong increase in minority absorption, $\mu_{\sigma-}$, is, therefore, due to a loss of minority states around $E_F$.

unoccupied $3d$ minority states. We envision that careful DOS-engineering in multi-component magnetic samples will allow to control the magnitude and direction of such ultrafast spin flow and therefore expect OISTR to play a decisive role for future ultrafast spintronics.

## Methods

**Theory and computational details.** The Runge-Gross theorem[42] establishes that the time-dependent external potential is a unique functional of the time-dependent density, given the initial state. Based on this theorem, a system of non-interacting particles can be chosen such that the density of this non-interacting system is equal to that of the interacting system for all times[43–45]. The wave function of this non-interacting system is represented as a Slater determinant of single-particle orbitals. In what follows a fully non-collinear spin-dependent version of these theorems is employed[15]. Then the time-dependent Kohn-Sham (KS) orbitals are Pauli spinors determined by the equations:

$$i\frac{\partial \psi_j(\mathbf{r}, t)}{\partial t} = \left[ \frac{1}{2}\left(-i\nabla + \frac{1}{c}\mathbf{A}_{ext}(t)\right)^2 + v_s(\mathbf{r}, t) \right.$$
$$\left. + \frac{1}{2c}\sigma \mathbf{B}_s(\mathbf{r}, t) + \frac{1}{4c^2}\sigma(\nabla v_s(\mathbf{r}, t) \times -i\nabla)\right]\psi_j(\mathbf{r}, t) \quad (1)$$

where $\mathbf{A}_{ext}(t)$ is a vector potential representing the applied laser field, and $\sigma$ are the Pauli matrices. The KS effective potential $v_s(\mathbf{r}, t) = v_{ext}(\mathbf{r}, t) + v_H(\mathbf{r}, t) + v_{xc}(\mathbf{r}, t)$ is decomposed into the external potential $v_{ext}$ ($\mathbf{r}$, $t$), the classical electrostatic Hartree potential $v_H$ ($\mathbf{r}$, $t$) and the exchange-correlation (XC) potential $v_{xc}$ ($\mathbf{r}$, $t$). Similarly, the KS magnetic field is written as $\mathbf{B}_s(\mathbf{r}, t) = \mathbf{B}_{ext}(t) + \mathbf{B}_{xc}(\mathbf{r}, t)$ where $\mathbf{B}_{ext}$ ($t$) is the magnetic field of the applied laser pulse plus possibly an additional magnetic field and $\mathbf{B}_{xc}$ ($\mathbf{r}$, $t$) is the XC magnetic field. The final term of Eq. (1) is the SO coupling term. All the implementations are done using the state-of-the art full potential linearized augmented plane wave (LAPW) method.

Time and energy resolved occupation of Kohn-Sham states, shown in Fig. 3b, can be calculated as follows:

$$A(\omega, t) = \sum_{i=1}^{\infty} \int_{BZ} \delta(\omega - \varepsilon_{i\mathbf{k}}) g_{i\mathbf{k}}(t), \text{with } g_{i\mathbf{k}}(t) = \sum_j n_{j\mathbf{k}} \left| O_{ij}^{\mathbf{k}}(t) \right|^2 \quad (2)$$

where $n_{j\mathbf{k}}$ is the occupation number of the $j^{th}$ time-evolving orbital, $\psi_j$ and

$$O_{ij}^{\mathbf{k}}(t) = \int d^3 r \phi_{i\mathbf{k}}^*(\mathbf{r}) \psi_{j\mathbf{k}}(\mathbf{r}, t). \quad (3)$$

Here $\phi_I$ are the ground-state Kohn-Sham orbitals. In absence of any time-dependent perturbation $\psi_{j\mathbf{k}}(\mathbf{r}, t = 0) = \phi_{j\mathbf{k}}(\mathbf{r})$ gives the ground-state DOS.

In the main text, we listed several mechanisms included in our simulations that can alter the magnetic moment. However, we wish to add that the much debated superdiffusive[15,16,32] spin currents are in principle implicitly included in the theoretical description[16], where the effect of electron-electron scattering is reproduced by the XC potential. However, to produce such currents, one would also require a prohibitively large supercell. This was not necessary in the present case, as our experiments explicitly exclude the contribution from superdiffusive

spin currents by probing the entire thickness of the magnetic sample in transmission geometry.

A fully non-collinear version of TDDFT as implemented within the Elk code[46] is used for all calculations. A regular mesh in **k**-space of 8×8×8 is used and a time step of $\Delta t = 0.1$ au is employed for the time-propagation algorithm[47]. A smearing width of 0.027 eV is used. Laser pulses used in the present work are linearly polarized (in-plane polarization) with a frequency of 1.55 eV, with a FWHM of 40 fs and incident fluence of 13 mJ/cm². The transient helicity dependent absorption is determined by performing linear-response calculations using the laser-excited state of the system at various times following the procedure detailed in[40].

During the time propagation, we see that there are small oscillations in the magnetization (with a large period) around the final value of the moment (see Fig. 2b). These oscillations are numerical and get damped as one increases the number of **k**-points. In contrast to this, the rapid oscillations are due to electrons moving back and forth with the frequency of the electric field (as well as higher harmonics). The local moments are extracted by integration of the magnetization within a sphere around each atom and this leads to a doubling of the frequency of any oscillation and hence the frequency in Fig. 2b is twice that of the pump-pulse frequency.

**Demagnetization time constants**. The magnetic asymmetry data, shown in Fig. 2, is described with a non-linear least square fit using the following formula:

$$A(t) = G(t) \otimes \Theta(t - t_0) \left[ A(t < t_0) + M_1 \left( 1 - e^{\frac{-t - t_0}{\tau_D}} \right) + M_2 \left( 1 - e^{\frac{-t - t_0}{\tau_R}} \right) \right] \quad (4)$$

where $\Theta(t - t_0)$ is the Heaviside function, $\tau_D$ and $\tau_R$ are the de- and remagnetization time constants and $t_0$ time delay zero, i.e., $t_0 = 0$. The function is convoluted with a Gaussian function, $G(t)$, with a full width half maximum of 60 fs, corresponding to the autocorrelation of the pump and probe pulses. The data was fitted between −2 ps < t < 2 ps to extract the correct time constants, as we observe an early onset of remagnetization, in particular in the Co film (see Supplementary Discussion as well as Supplementary Fig. 1 and Table 1 for more details). All stated errors correspond to an uncertainty interval of σ (68%).

## Data availability

The data that supports the finding of this study are available from the corresponding author upon reasonable request.

## Code availability

The Elk code is released under GPL and is freely available. It can be downloaded from the webpage: elk.sourceforge.net.

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

## Acknowledgements

S.S., C.v.K.S., and S.E. would like to thank DFG for funding through TRR227 projects A02 and A04.

## Author contributions

S.E., S.S., F.W. and C.v.K.S. conceived the experiment, S.S., P.E. and J.K.D performed the ab initio calculations. F.W., D.S., C.S. and C.v.K.S. performed the experiment. D.E. grew the magnetic samples. F.W. analysed the experimental data. C.v.K.S., F.W. and S.S. wrote the manuscript with discussions and improvements from all authors.

## Competing interests

The authors declare no competing interests.
