## [Peer Review File · Nature Communications]

Reviewers' comments:

Reviewer #1 (Remarks to the Author):

The manuscript by Willems et al presents a combined experiment/theory study to demonstrate optically induced spin transfer from Pt to Co valence levels in ferromagnetic alloys. Although such a demonstration would undoubtedly be of interest for the readers of Nature Communications, the paper falls short in its claim as I will describe in more detail below:

1. Experimentally, element-specific magnetometry at shallow core-valence resonances is used to study Co and Pt magnetic moment evolution for two materials, Co and CoPt. The main observation is an identical demagnetization of Co and Pt in CoPt but a significantly slower one in Co. The difference between Co and CoPt demagnetizations has been observed before (Kuiper 2014).
2. The authors use the identical Co and Pt demagnetization and claim that (line 129/130) 'a merely SO (spin-orbit coupling) mediated demagnetization process would entail a faster demagnetization for Pt'. Presumably, this unproven statement is meant to exclude the commonly accepted demagnetization process via inelastic coupling to phonons (Koopmans et al. Nature Materials 9, 259 (2010)). I note that this process is neither properly discussed or referenced nor included in the theoretical modeling.
3. Modeling is limited to the theory used for predicting the optically induced spin transfer. Unfortunately the strengths and weaknesses of this approach are not properly discussed in the present manuscript. Reading through the references it becomes clear that demagnetization via energy dissipation to phonons is not included. Neither is the well-known spin-dependent electronic scattering (Aeschlimann et al PRL 79, 5158 (1997)) that ultimately leads to the so-called superdiffusive spin currents (Battiato 2010), another energy dissipation mechanism extensively discussed in ultrafast magnetism. In my opinion, omission of such processes severely limits the predictive power of the modeling.
4. Fig. 3 contains a comparison of helicity dependent Co x-ray absorption spectroscopy for Co and CoPt to the calculated spin-dependent valence level population dynamics. The authors claim that the qualitative agreements of experimental and theoretical curves prove the proposed optical spin transfer. This, however, remains wishful thinking unless they can credibly demonstrate that other demagnetization processes discussed in the literature have different behaviors.
5. Why are no data for the Pt magnetization dynamics included in Fig. 3? The authors claim theoretically that there is an optical spin transfer from Pt to Co. What is its influence on the spin-resolved Co AND Pt valence level population? Is this reflected in the experiment? Directly comparing Co and Pt XAS and XMCD is crucial for the paper's claim.
6. Why are not all fitting parameters included in Tab. 1? In addition the table seems to contain only Co magnetic moment results. I would also like to see what is happening for the Pt magnetic moments in CoPt. This selective presentation makes a critical verification of the claims of this paper difficult.

In conclusion, I cannot recommend publication of the manuscript in its present form as it contains a too selective presentation of experimental and theoretical results. In addition, an open minded discussion of alternative scenarios needs to be included in any resubmission

Reviewer #2 (Remarks to the Author):

This paper reports that the ultrafast demagnetization in CoPt is faster than in pure Co and assigns this to optically induced spin transfer (OIST) from Pt to Co, in particular minority electrons from Pt 5d to

Co 3d. This conjecture is supported by XMCD experiments and theoretical calculations. The XMCD experiments uses Higher Harmonic Generation to exploit 60 eV and 54 eV photons to probe the minority/majority spins in the 3d and 5d states of Co and Pt, respectively. The data and calculations are quite clearly showing that the demagnetization in CoPt is faster than in Co and the authors suggest a new mechanism of OIST. These results clearly shine new light on the intricate mechanisms of ultrafast spin and magnetisation dynamics and will be of strong interest to the still expanding field. As such the paper is certainly suitable for NC. However, before it can be published, the authors should address a number of points:

1) the abstract mentions that the E-field of the laser pulse transfers minority electrons from Pt 5d to Co 3d. However, in the rest of the paper this is not mentioned, and rightly so I think. It is an optical excitation that can bring electrons from the 5d to the 3d states, the em-field itself is an ac-field! (~1014 Hz).

2) Fig.2 shows the time dependence of the relative magnetic asymmetry for both Co and Pt after optical excitation with a 60fs, 800nm pulse, showing that the demagnetization is faster in the CoPt alloy than in pure Co. The calculations based on their suggested OIST (for the first 75fs) agree with this.

What are the ~1fs period oscillations in the calculations (Fig.2b:)? And the slower, ~20-25fs, period? Please explain.

3) In Fig.3: The helicity dependent absorption for Co($\mu +$) increases immediately after the pump pulse, and at 70fs is more or less stable (at 1.01); the Co($\mu -$) does start to decrease only after 70fs; in CoPt, both Co($\mu +$) and Co($\mu -$) immediately start to change with pump pulse, the decrease in Co($\mu -$) is also stronger than for the pure Co. It is not clear to this referee why this is the case nor is it explained in the paper.

4) The inset depicts an integration over all energies of occupied states, n , and shows that in CoPt, the increase of minority electrons is not matched by a loss of majority electrons. This is explained by OIST transferring minority electrons from Pt to Co. Actually, the experiment shows that there is an asymmetry in the loss of majority and gain in minority electrons, but it does not prove that these electrons come from Pt 5d, THIS IS A CONJECTURE.

Though this may be a possible explanation, I do not think this is hard evidence. I suggest that the authors state this a bit more careful.

as a minor comment: In Fig.3: please add "normalized helicity dependent absorption, μ_{\pm} " in the caption

5) In the discussion the authors make a mistake I think: LINE 191/192: We observe a symmetric response for CoPt, with an increase in majority electrons and a balanced decrease in minority electrons. THIS SOUNDS LIKE THE OPPOSITE OF WHAT WAS SAID BEFORE IN LINE 163/164, and also against the message of the paper: in Co the response is symmetric, but in CoPt it is not, because of the proposed Pt contribution.

6) Why are the timescales for Co and Pt in CoPt the same? Line 207-209: "However, a locally enhanced spin-orbit coupling on Pt leads to more spin-flips causing a loss in the moment. In other words, we observe a competition of two processes, namely OISTR and spin-flips, leading to similar

dynamics in the MCD(t) signal for Co and Pt in the CoPt alloy". That would be a remarkable coincidence, so I don't think this is a very convincing statement.

We thank the referees for their constructive criticism and helpful comments. The referees have raised important issues, which we have addressed in detail below.

Response to the report of Reviewer #1:

The manuscript by Willems et al presents a combined experiment/theory study to demonstrate optically induced spin transfer from Pt to Co valence levels in ferromagnetic alloys. Although such a demonstration would undoubtedly be of interest for the readers of Nature Communications, the paper falls short in its claim as I will describe in more detail below:

1. Experimentally, element-specific magnetometry at shallow core-valence resonances is used to study Co and Pt magnetic moment evolution for two materials, Co and CoPt. The main observation is an identical demagnetization of Co and Pt in CoPt but a significantly slower one in Co. The difference between Co and CoPt demagnetizations has been observed before (Kuiper 2014).

2. The authors use the identical Co and Pt demagnetization and claim that (line 129/130) ‘a merely SO (spin-orbit coupling) mediated demagnetization process would entail a faster demagnetization for Pt’. Presumably, this unproven statement is meant to exclude the commonly accepted demagnetization process via inelastic coupling to phonons (Koopmans et al. Nature Materials 9, 259 (2010)). I note that this process is neither properly discussed or referenced nor included in the theoretical modeling.

The referee has raised two important issues: (a) whether the momentum flow from spins to phonons (i.e., nuclear degrees of freedom) is taken into account in the theoretical description and (b) the nature of SO coupling-induced demagnetization. In the following, we answer these concerns in detail. Furthermore, motivated by the referees’ concerns, we have now added these details also to the manuscript (starting line 111ff and 144ff)

(a) Following mechanisms have been included in the theoretical study: (i) spin and charge currents due to laser excitation; this term is responsible for the OISTR mechanism, (ii) spin-orbit induced spin-flips, (iii) effect of these spin-flips on the spin-currents (Krieger et al. 2017) (iv) electron-electron scattering effects via the exchange-correlation functional, and (v) coupling of the electrons to nuclear degrees of freedom; the change in charge density as a function of time (due to laser excitation) leads to large forces on the nuclei which are coupled to the spins via Coulomb and spin-orbit terms. This coupling implies that the momentum is allowed to flow from spin and orbital degrees of freedom to nuclei. Second-order effects are missing in our simulations-- these forces displace nuclei (generate phonons), and this, in turn, causes charge and spin densities to change. However, we note that such second-order effects start to cause significant changes to the magnetization only after ~100fs, and hence we have focused upon the early time physics in the present work.

(b) The spin-orbit coupling term in the Hamiltonian (see Eq. 1 in the supplementary information) is proportional to the gradient of the local-potential. This implies that the effect of the SO term is most prominent for localized electrons where the gradient term is large. Thus localized electrons are more prone to flipping of the spins due to the SO term (we demonstrated this explicitly in (Krieger et al. 2015)). This implies that the SO induced spin-flips would show very different rates for the two species with largely different SO coupling strengths (SO for Pt is ~10 times larger than for Co (Shanavas, Popović, and Satpathy 2014)). However, a flow of a current (i.e. OISTR) between the two species would lead to a mixing of the demagnetization times. In the present work, we demonstrate this both experimentally, due to the element specificity of our technique, and theoretically.

It is also important to mention that this is not orthogonal but instead in line with the work of (Koopmans et al. 2010). In the microscopic three-temperature model a spin-flip probability, \square_{sf} , is introduced, which is proportional to the spin-mixing and is usually considered to scale with the nuclear charge Z (it can be shown that for outer electrons it scales according to Z^2 (Shanavas, Popović, and

Satpathy 2014) because of spin-orbit coupling. In other words, a higher spin-orbit coupling entails a larger demagnetization rate.

3. Modeling is limited to the theory used for predicting the optically induced spin transfer. Unfortunately the strengths and weaknesses of this approach are not properly discussed in the present manuscript. Reading through the references it becomes clear that demagnetization via energy dissipation to phonons is not included. Neither is the well-known spin-dependent electronic scattering (Aeschlimann et al PRL 79, 5158 (1997)) that ultimately leads to the so-called superdiffusive spin currents (Battiato 2010), another energy dissipation mechanism extensively discussed in ultrafast magnetism. In my opinion, omission of such processes severely limits the predictive power of the modeling.

It is important to mention that OISTR is not the only process included in the theory, but rather it emerges in our simulations as one of the most important processes in the early times ($t < 100$ fs). We have now clearly stated this in the manuscript (lines 111ff) together with the list of all the other processes that are included in the simulations (please see response to the previous point for a complete list).

As for the superdiffusive currents, they are in principle included in the theoretical description (Krieger et al. 2015) where the effect of electron-electron scattering is reproduced by the XC potential, but for its complete description one would require a super-cell. However, such simulations were not needed in the present case because, in our experiments, we explicitly exclude the contribution from superdiffusive spin currents by probing the entire thickness of the magnetic sample in transmission geometry and because our samples are grown on insulating membranes (Si₃N₄ or oxidized Al). This is now explicitly mentioned in our manuscript (line 92ff and 258ff in SI).

4. Fig. 3 contains a comparison of helicity dependent Co x-ray absorption spectroscopy for Co and CoPt to the calculated spin-dependent valence level population dynamics. The authors claim that the qualitative agreements of experimental and theoretical curves prove the proposed optical spin transfer. This, however, remains wishful thinking unless they can credibly demonstrate that other demagnetization processes discussed in the literature have different behaviors.

Addressing the Referees' concerns, we have now clearly stated in the manuscript that OISTR *emerges* as a dominant mechanism among other important processes discussed in literature. In particular, our simulations take into account electron-electron scattering, momentum transfer to phonons, spin-currents and effects of SO coupling on spin-currents.

Most importantly, we have added two extra sets of data to strengthen our analysis:

(a) The comparison between experiment and theory is now performed by directly calculating the response functions theoretically, i.e., μ_{\pm} . The good agreement between theory and experiment further strengthens our observation of spin-selective charge transfer from Pt to Co *d* states.

(b) To further corroborate this, we have also added the experimental data for helicity dependent absorption at Pt N₇-edge (Fig. 2s in SI). This data gives a clear indication of a loss of Pt minority charge.

5. Why are no data for the Pt magnetization dynamics included in Fig. 3? The authors claim theoretically that there is an optical spin transfer from Pt to Co. What is its influence on the spin-resolved Co AND Pt valence level population? Is this reflected in the experiment? Directly comparing Co and Pt XAS and XMCD is crucial for the paper's claim.

As regards the referees' criticism, we have now included additional data on Pt to further corroborate our main hypothesis. In Fig. 2, we now also show the relative asymmetry $A(t)/A(t < 0)$ measured at the

Pt N₇ edge at 72.6 eV. This data, together with the data measured at 54.1 eV, demonstrates that the magnetic moment of Pt is reduced on the same time scale as the Co moment measured at 60.3 eV. Additionally, we also now show the helicity dependent absorption around the Pt N₇ edge in the SI (line 317ff and Fig 2s). The observed loss of minority electrons at the Pt atom is interpreted as the source for the efficient filling of minority states in Co. Finally, we now also show the calculated total number of occupied electrons (minority and majority electrons integrated over the energy range from -13 eV to 13 eV) as a function of time in Fig 1s in the SI (line 306ff). While we see no change for the Co film, i.e., spin-flips result in a loss of majority electrons equal to a gain in minority electrons because of charge conservation, the situation is clearly different in the CoPt alloy. For Co we see an increase of minority electrons in excess to the loss of majority electrons (compare inset of Fig. 3c) and hence a total increase of the integrated occupations. In contrast, in Pt, the loss in majority electrons due to efficient spin-flip processes does not correspond to the gain in minority electrons; the total occupation drops as minority electrons are efficiently transferred from Pt to empty Co states.

6. Why are not all fitting parameters included in Tab. 1? In addition the table seems to contain only Co magnetic moment results. I would also like to see what is happening for the Pt magnetic moments in CoPt. This selective presentation makes a critical verification of the claims of this paper difficult.

We have now included all relevant fitting parameters and their corresponding 1 σ confidence intervals in Table 1s. These include the static magnetic asymmetry $A(t<0)$, the relative demagnetization amplitude $M_1/A(t<0)$ as well as the demagnetization time constants, τ_D . Furthermore, we have now stated more clearly that the table does include the response of both elements, Co and Pt for the CoPt alloy. At 54.1 eV and 57.1 eV we predominantly probe the Pt O₃ resonance (Pt 5p_{3/2} \square Pt 5d), and at 72.6 eV, we probe the Pt N₇ resonance (Pt 4f_{5/2,7/2} \square Pt 5d). In other words, at these three photon energies, we follow the magnetic moment of Pt. At 60.3 eV, we predominantly probe the Co M_{2,3} resonances (3p_{1/2,3/2} \square 3d) and probe the Co magnetic moment. At other energies, the resonances overlap, and we have contributions from both elements. We have also extended the discussion of Table 1 to emphasize the important statement that, within our experimental error, $M_1/A(t<0)$ and τ_D show no element specificity (see line 290ff).

In conclusion, I cannot recommend publication of the manuscript in its present form as it contains a too selective presentation of experimental and theoretical results. In addition, an open minded discussion of alternative scenarios needs to be included in any resubmission

We have now improved the manuscript by (i) adding an extra data set to support our line of argument – we now directly compare the measured response functions (helicity dependent absorption) with theoretical calculations of exactly the same observable. This corroborates our claim that helicity dependent absorption probes relative changes in the spin-split electron occupations around the Fermi level. (ii) We have expanded our discussion of the processes which are included in the *ab-initio* description and (iii) have shown additional experimental and theoretical data of the charge dynamics of Pt, further strengthening our hypothesis that the OISTR mechanism dominates the early time spin dynamics.

We hope this will satisfy Referees concerns

We thank the referee for his/her constructive criticism and helpful comments. The referee has raised important issues, which we have addressed in detail below.

Response to the report of Reviewer #2

This paper reports that the ultrafast demagnetization in CoPt is faster than in pure Co and assigns this to optically induced spin transfer (OIST) from Pt to Co, in particular minority electrons from Pt 5d to Co 3d. This conjecture is supported by XMCD experiments and theoretical calculations. The XMCD experiments uses Higher Harmonic Generation to exploit 60 eV and 54 eV photons to probe the minority/majority spins in the 3d and 5d states of Co and Pt, respectively. The data and calculations are quite clearly showing that the demagnetization in CoPt is faster than in Co and the authors suggest a new mechanism of OIST. These results clearly shine new light on the intricate mechanisms of ultrafast spin and magnetisation dynamics and will be of strong interest to the still expanding field. As such the paper is certainly suitable for NC. However, before it can be published, the authors should address a number of points:

1) the abstract mentions that the E-field of the laser pulse transfers minority electrons from Pt 5d to Co 3d. However, in the rest of the paper this is not mentioned, and rightly so I think. It is an optical excitation that can bring electrons from the 5d to the 3d states, the em-field itself is an ac-field! (~1014 Hz).

We have deleted the words in the abstract “in conjunction with the electric field” as we agree that this is misleading and is not important for what follows.

2) Fig.2 shows the time dependence of the relative magnetic asymmetry for both Co and Pt after optical excitation with a 60fs, 800nm pulse, showing that the demagnetization is faster in the CoPt alloy than in pure Co. The calculations based on their suggested OIST (for the first 75fs) agree with this.

What are the ~1fs period oscillations in the calculations (Fig.2b:)? And the slower, ~20-25fs, period? Please explain.

We explain the origin of the oscillation in the SI (starting line 274)

During the time propagation, we see that there are small oscillations (with large period) around the final value of the moment (see Fig. 2b of the manuscript). These oscillations are numerical and get damped as you increase the number of k-points. In contrast to this, the rapid oscillations are due to the electrons moving back and forth with the frequency of the electric field (as well as higher harmonics). The local moments are extracted by integration of the magnetization within a sphere around each atom, and this leads to a doubling of the frequency of any oscillation and hence the frequency in Figs. 2(b) of the manuscript is twice that of the pump-pulse frequency (~2.7 fs).

In this context it may be noteworthy, that Fig. 2 compares the measured magnetic asymmetry, $A(t)$, to the magnetic moment, $m(t)$, i.e. not the identical observables. (This is in contrast to Fig. 3, where we directly compare the response function, the observable μ). Since MCD predominantly probes localized

d-bands, which reside within the integration sphere mentioned above, it will be challenging to observe these fast oscillations with our experimental approach. Here, other experimental techniques may be more appropriate.

3) In Fig.3: The helicity dependent absorption for $\text{Co}(\mu +)$ increases immediately after the pump pulse, and at 70fs is more or less stable (at 1.01); the $\text{Co}(\mu -)$ does start to decrease only after 70fs; in CoPt, both $\text{Co}(\mu +)$ and $\text{Co}(\mu -)$ immediately start to change with pump pulse, the decrease in $\text{Co}(\mu -)$ is also stronger than for the pure Co. It is not clear to this referee why this is the case nor is it explained in the paper.

We have considerably improved Figure 3 by now directly comparing the measured to calculated helicity dependent absorption, i.e., we now directly compare the same response functions. Note that the details of the absorption curves are very well reproduced and hence can be directly attributed to changes in transient DOS occupations.

$\text{Co}(\mu +)$ (directly related to changes of majority occupations) increases and stays constant because majority electrons are lost during the demagnetization in spin-flip processes. We mention this now explicitly (starting line 182ff)

The processes responsible for the evolution of μ_{\square} (reflecting changes in the minority occupations) are described in detail starting in line 194ff. In particular, we now also say explicitly:

Only after about 60 fs with progressing electron thermalization spin-flip processes, i.e., an increase of minority electrons, are directly evidenced by a decrease of μ_{\square} .

4) The inset depicts an integration over all energies of occupied states, n , and shows that in CoPt, the increase of minority electrons is not matched by a loss of majority electrons. This is explained by OIST transferring minority electrons from Pt to Co. Actually, the experiment shows that there is an asymmetry in the loss of majority and gain in minority electrons, but it does not prove that these electrons come from Pt 5d, THIS IS A CONJECTURE.

Though this may be a possible explanation, I do not think this is hard evidence. I suggest that the authors state this a bit more careful.

We have rewritten this section of the paper (line 194ff) and have given additional experimental evidence by showing data for the Pt N_7 edge in the SI.

The large increase in the number of minority spin electrons local to Co between the pure Co and the CoPt alloy must be due to optical excitation of electrons in occupied bands below the Fermi level. When Pt is added, we see from the DOS shown in Fig. 1b that these are Pt 5d states. We have now further strengthened this argument by comparing the total integrated occupations of Co as well as of Co and Pt in CoPt in Fig. 1s in the supplementary information. While we see no change for the Co film, i.e., spin-flips result in a loss of majority electrons equal to a gain in minority electrons because of charge conservation, the situation is clearly different in the CoPt alloy. For Co we see an increase of minority electrons in excess to the loss of majority electrons (compare inset of Fig. 3c) and hence a total increase of the integrated occupations. In contrast, in Pt the loss of majority electrons due to efficient spin-flip processes is not equal to the gain in minority electrons; the total occupation decreases as minority electrons are efficiently transferred from Pt to empty Co states.

as a minor comment: In Fig.3: please add "normalized helicity dependent absorption, μ_{\pm} " in the caption

We have added this in the caption of Fig. 3.

5) In the discussion the authors make a mistake I think: LINE 191/192: We observe a symmetric response for CoPt, with an increase in majority electrons and a balanced decrease in minority electrons. THIS SOUNDS LIKE THE OPPOSITE OF WHAT WAS SAID BEFORE IN LINE 163/164, and also against the message of the paper: in Co the response is symmetric, but in CoPt it is not, because of the proposed Pt contribution.

We agree with the referee that we have used an unfortunate phrasing by using the word “symmetric” in the sentence in lines 191/192. What we meant by this is the fact that in case of CoPt the two response functions (μ_{\pm}) are mirrors of each other in that as one increases the other correspondingly decreases, i.e., they look “symmetric” around 0. This is in total contrast to the response functions in Co where one increases while the second stays constant, i.e., they look asymmetric.

We thank the Referee for pointing this out. We have removed figure 3c (integrated DOS), but instead show the actual calculated response function, μ_{\pm} , in Figure 3b, and have completely rewritten this part of the manuscript thereby removing this ambiguity. The former Figure 3b became now Figure 3c showing the calculated changes in the spin resolved density of states at a selected time ($t=41$ fs).

Starting from line 201 we now discuss the inset of Fig 3 (c), showing the number of occupied spin-split d -states, n_d , integrated in the energy range -13 eV to +13 eV. The essential point here is that from the analysis of the *total* occupation, we observe a net increase of occupation on the Co sites in CoPt which *must* originate from the transfer of spins from Pt to Co. In contrast, in the pure Co the net occupation is unchanged as one would expect due to charge conservation. We additionally discuss this observation in more detail starting from line 306 and Fig 1s.

6) Why are the timescales for Co and Pt in CoPt the same? Line 207-209: "However, a locally enhanced spin-orbit coupling on Pt leads to more spin-flips causing a loss in the moment. In other words, we observe a competition of two processes, namely OISTR and spin-flips, leading to similar dynamics in the MCD(t) signal for Co and Pt in the CoPt alloy". That would be a remarkable coincidence, so I don't think this is a very convincing statement.

As regards the referees' criticism, we have greatly extended the discussion of the two processes responsible for demagnetization in CoPt. The important point in our line of argument is that because of larger spin-orbit coupling for Pt and the fact that SO is a local property - the definition of SO coupling (last term in Eq. 1 in SI) being proportional to the gradient of the local potential - implies that SO coupling is an intrinsic local property and, therefore, distinct for atoms with different SO. This implies that Pt should demagnetize substantially faster than Co. In contrast to this, we observed equal demagnetization times for Co and Pt in the CoPt alloy, both in experiment and theory.

At this point, it is important to mention that, so far, comparison between Co and CoPt samples have relied on all-optical spectroscopy where only the global response of the whole system was probed and CoPt could only be described by a single average spin-flip scattering constant (Kuiper et al. 2014). Due to the element specific nature of our experiments and theory we can now say that the element-resolved demagnetization times for both Co and Pt in CoPt are faster than for pure Co.

To understand this surprising observation, we probed deeper and found that we see a striking difference in the minority absorption between Co and CoPt at Co edge, i.e., an efficient filling of Co minority states. This behavior evidences a second process, which is not driven by spin-flips. Despite including processes like SO mediated spin-flips, angular momentum transfer to nuclei, spin-current and changes in spin-currents due to SO in the calculation (see line 111ff for the full list), we find that OISTR emerges as the second dominant mechanism leading to this striking difference between Co in CoPt and pure Co films. Furthermore, we now corroborate this theoretical finding by looking at the Pt response and in doing so clearly see a charge transfer from Pt minority states to Co minority states (see Fig. 1S in SI). This discussion is now much more detailed (line 144ff) in the manuscript.

- Koopmans, B. et al. 2010. "Explaining the Paradoxical Diversity of Ultrafast Laser-Induced Demagnetization." *Nature Materials* 9(3): 259–65.
- Krieger, K. et al. 2015. "Laser-Induced Demagnetization at Ultrashort Time Scales: Predictions of TDDFT." *Journal of Chemical Theory and Computation* 11(10): 4870–74.
- Krieger, K. et al. 2017. "Ultrafast Demagnetization in Bulk versus Thin Films: An Ab Initio Study." *Journal of Physics: Condensed Matter* 29(22): 224001.
- Kuiper, K C et al. 2014. "Spin-Orbit Enhanced Demagnetization Rate in Co/Pt-Multilayers." *Applied Physics Letters* 105(20): 202402.
- Shanavas, K V, Z. S. Popović, and S. Satpathy. 2014. "Theoretical Model for Rashba Spin-Orbit Interaction in d Electrons." *Physical Review B* 90(16): 165108.

REVIEWERS' COMMENTS:

Reviewer #1 (Remarks to the Author):

In the revised version of their manuscript the authors have addressed all of my comments satisfactorily and I support publication in Nature Communications. In particular I appreciate the addition of Fig. 2s which together with the CoPt data in Fig. 3a demonstrates charge transfer from Pt to Co, the central point of the paper. I also appreciate the discussion of the theory and its current limitation. However, I urge the authors to consider moving Fig. 2s to the main text. In my opinion it is the experimental data that will serve the scientific community as a highly visible benchmark for developing more complete theoretical approaches beyond the first 100fs. For this reason it would also be good if all experimental data were shown over the whole (or at least the same) measured temporal range and not limited to times where the current theory works. Finally the authors should consider including the demagnetization data. Without it the formula in line 292f describing demagnetization and remagnetization is incomplete. Either remagnetization affects the fitted demagnetization timescales in which case the data and fit parameters should be included in the manuscript or if not then the last term in the formula can be omitted.

Reviewer #2 (Remarks to the Author):

I have read the response of the authors to the referees comments and I am satisfied with them. As far as I am concerned, the paper can be published in the present form.

Response to issues raised by referees.

Reviewer #1 (Remarks to the Author):

In the revised version of their manuscript the authors have addressed all of my comments satisfactorily and I support publication in Nature Communications. In particular I appreciate the addition of Fig. 2s which together with the CoPt data in Fig. 3a demonstrates charge transfer from Pt to Co, the central point of the paper. I also appreciate the discussion of the theory and its current limitation. However, I urge the authors to consider moving Fig. 2s to the main text. In my opinion it is the experimental data that will serve the scientific community as a highly visible benchmark for developing more complete theoretical approaches beyond the first 100fs. For this reason it would also be good if all experimental data were shown over the whole (or at least the same) measured temporal range and not limited to times where the current theory works. Finally the authors should consider including the demagnetization data. Without it the formula in line 292f describing demagnetization and remagnetization is incomplete. Either remagnetization affects the fitted demagnetization timescales in which case the data and fit parameters should be included in the manuscript or if not then the last term in the formula can be omitted.

Our response:

We have followed the suggested improvements from reviewer #1 as follows:

1. Figure 2 of the supplement showing the helicity dependent measurement at the Pt N_7 is now shifted to the main text and is now called Figure 4. We have slightly modified the describing text to accommodate this change.
2. As suggested, we now show the magnetic asymmetry, $A(t)$, in a longer time interval, beyond where TDDFT calculations are currently valid. This is now Fig. 1 in the supplementary information. We now explicitly mention that indeed the data has to be described by a double-exponential function to extract the correct time constants, as we observe an early onset of remagnetization, in particular in the Co film.

Reviewer #2 (Remarks to the Author):

I have read the response of the authors to the referees comments and I am satisfied with them. As far as I am concerned, the paper can be published in the present form.